# Detect Rumors in Microblog Posts Using Propagation Structure via Kernel Learning

## Abstract

How fake news goes viral via social media? How does its propagation pattern differ from real stories? In this paper, we attempt to address the problem of identifying rumors, i.e., fake information, out of microblog posts based on their propagation structure. We firstly model microblog posts diffusion with propagation trees, which provide valuable clues on how an original message is transmitted and developed over time. We then propose a kernel-based method called Propagation Tree Kernel, which captures high-order patterns differentiating different types of rumors by evaluating the similarities between their propagation tree structures. Experimental results on two real-world datasets demonstrate that the proposed kernel-based approach can detect rumors more quickly and accurately than state-of-the-art rumor detection models.

## 1 Introduction

On November 9th, 2016, Eric Tucker, a grassroots user who had just about 40 followers on Twitter, tweeted his unverified observations about paid protesters being bused to attend anti-Trump demonstration in Austin, Texas. The tweet, which was proved false later, was shared over 16 thousand times on Twitter and 350 thousand times on Facebook within a couple of days, fueling a nation-wide conspiracy theory[1]. The diffusion of the story is illustrated as Figure 1 which gives the key spreading points of the story along the time line. We can see that after the initial post, the tweet

---

[1] https://www.nytimes.com/2016/11/20/
business/media/how-fake-news-spreads.
html

was shared or promoted by some influential online communities and users (including Trump himself), resulting in its wide spread.

A widely accepted definition of rumor is "unverified and instrumentally relevant information statements in circulation" (DiFonzo and Bordia, 2007). This unverified information may eventually turn out to be true, or partly or entirely false. In today's ever-connected world, rumors can arise and spread at lightening speed thanks to social media platforms, which could not only be wrong, but be misleading and dangerous to the public society. Therefore, it is crucial to track and debunk such rumors in timely manner.

Journalists and fact-checking websites such as `snopes.com` have made efforts to track and detect rumors. However, such endeavor is manual, thus prone to poor coverage and low speed. Feature-based methods (Castillo et al., 2011; Yang et al., 2012; Ma et al., 2015) achieved certain success by employing large feature sets crafted from message contents, user profiles and holistic statistics of diffusion patterns (e.g., number of retweets, propagation time, etc.). But such an approach was over simplified as they ignored the dynamics of rumor propagation. Existing studies considering propagation characteristics mainly focused on the temporal features (Kwon et al., 2013, 2017) rather than the structure of propagation.

So, can the propagation structure make any difference for differentiating rumors from non-rumors? Recent studies showed that rumor spreaders are persons who want to get attention and popularity (Sunstein, 2014). However, popular users who get more attention on Twitter (e.g., with more followers) are actually less likely to spread rumor in a sense that the high audience size might hinder a user from participating in propagating unverified information (Kwon et al., 2017). Intuitively, for "successful" rumors being propagated as widely

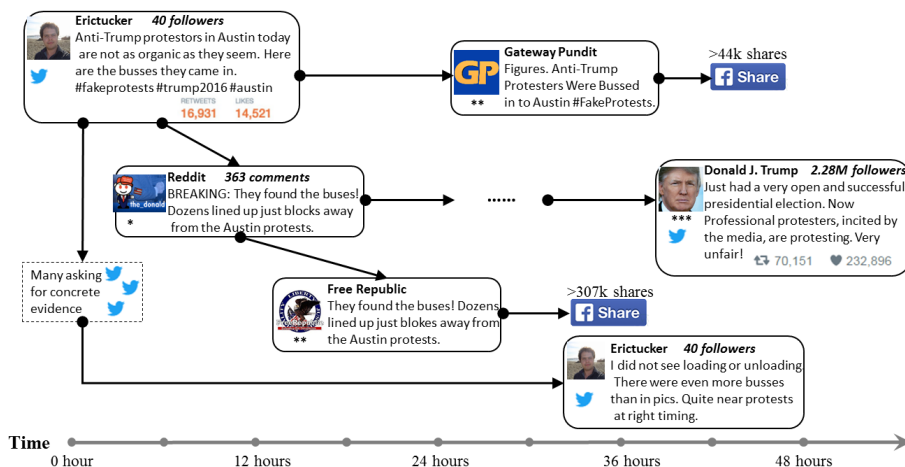

Figure 1: An illustration of how the rumor about "buses used to ship in paid anti-Trump protesters to Austin, Texas" becomes viral, where '*' indicates the level of influence.

as popular real news, initial spreaders (typically lack of popularity) must attract certain amount of broadcasting power, e.g., attention of influential users or communities that have a lot of audiences joining in promoting the propagation. We refer to this as a constrained mode propagation, relative to the open mode propagation of normal messages that everyone is open to share. Such different modes of propagation may imply some distinct propagation structures between rumors and non-rumors and even among different types of rumors.

Due to the complex nature of information diffusion, explicitly defining discriminant features based on propagation structure is difficult and biased. Figure 2 exemplifies the propagation structures of two Twitter posts, a rumor and a non-rumor, initiated by two users shown as the root nodes (in green color). The information flows here illustrates that the rumorous tweet is first posted by a low-impact user, then some popular users joining in who boost the spreading, but the non-rumorous tweet is initially posted by a popular user and directly spread by many general users; content-based signal like various users' stance (Zhao et al., 2015) and edge-based signal such as relative influence (Kwon et al., 2017) can also suggest the different nature of source tweets. Many of such implicit distinctions throughout propagation are hard to hand craft specifically using flat summary of statistics as previous work did. In addition, unlike representation learning for plain text, learning for representation of structures such as networks is not well studied in general. Therefore, traditional and latest text-based models (Castillo et al., 2011;

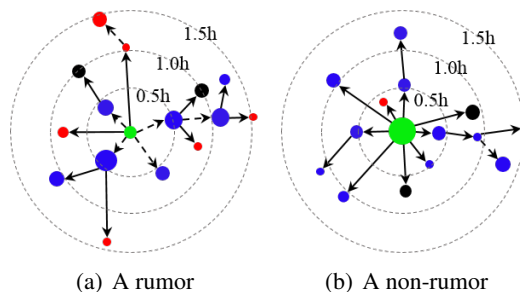

(a) A rumor   (b) A non-rumor

Figure 2: Fragments of the propagation for two source tweets. Node size: denotes the popularity of the user who tweet the post (represented by # of followers); Red, black, blue node: content-wise the user express doubt/denial, support, neutrality in the tweet, respectively; Solid (dotted) edge: information flow from a more (less) popular user to a less (more) popular user; Dashed concentric circles: time stamps.

Ma et al., 2015, 2016) cannot be applied easily on such complex, dynamic structures.

To capture high-order propagation patterns for rumor detection, we firstly represent the propagation of each source tweet with a propagation tree which is formed by harvesting user's interactions to one another triggered by the source tweet. Then, we propose a kernel-based data-driven method called Propagation Tree Kernel (PTK) to generate relevant features (i.e., subtrees) automatically for estimating the similarity between two propagation trees. Unlike traditional tree kernel (Moschitti, 2006; Zhang et al., 2008) for modeling syntactic structure based on parse tree, our propagation tree consists of nodes corresponding to microblog

posts, each represented as a continuous vector, and edges representing the direction of propagation and providing the context to individual posts. The basic idea is to find and capture the salient substructures in the propagation trees indicative of rumors. We also extend PTK into a context-enriched PTK (cPTK) to enhance the model by considering different propagation paths from source tweet to the roots of subtrees, which capture the context of transmission. Extensive experiments on two real-world Twitter datasets show that the proposed methods outperform state-of-the-art rumor detection models with large margin.

Moreover, most existing approaches regard rumor detection as a binary classification problem, which predicts a candidate hypothesis as rumor or not. Since a rumor often begins as unverified and later turns out to be confirmed as true or false, or remains unverified (Zubiaga et al., 2016), here we consider a set of more practical, finer-grained classes: false rumor, true rumor, unverified rumor, and non-rumor, which becomes an even more challenging problem.

## 2 Related Work

Tracking misinformation or debunking rumors has been a hot research topic in multiple disciplines (DiFonzo and Bordia, 2007; Morris et al., 2012; Rosnow, 1991). Castillo et al. (2011) studied information credibility on Twitter using a wide range of hand-crafted features. Following that, various features corresponding to message contents, user profiles and statistics of propagation patterns were proposed in many studies (Yang et al., 2012; Wu et al., 2015; Sun et al., 2013; Liu et al., 2015). Zhao et al. (2015) focused on early rumor detection by using regular expressions for finding questing and denying tweets as the key for debunking rumor. All such approaches are over simplistic because they ignore the dynamic propagation patterns given the rich structures of social media data.

Some studies focus on finding temporal patterns for understanding rumor diffusion. Kown et al. (2013; 2017) introduced a time-series fitting model based on the temporal properties of tweet volume. Ma et al. (2015) extended the model using time series to capture the variation of features over time. Friggeri et al. (2014) and Hannak et al. (2014) studied the structure of misinformation cascades by analyzing comments linking to rumor debunking websites. More recently, Ma et al. (2016) used recurrent neural networks to learn the representations of rumor signals from tweet text at different times. Our work will consider temporal, structural and linguistic signals in a unified framework based on propagation tree kernel.

Most previous work formulated the task as classification at event level where an event is comprised of a number of source tweets, each being associated with a group of retweets and replies. Here we focus on classifying a given source tweet regarding a claim which is a finer-grained task. Similar setting was also considered in (Wu et al., 2015; Qazvinian et al., 2011).

Kernel methods are designed to evaluate the similarity between two objects, and tree kernel specifically addresses structured data which has been successfully applied for modeling syntactic information in many natural language tasks such as syntactic parsing (Collins and Duffy, 2001), question-answering (Moschitti, 2006), semantic analysis (Moschitti, 2004), relation extraction (Zhang et al., 2008) and machine translation (Sun et al., 2010). These kernels are not suitable for modeling the social media propagation structures because the nodes are not given as discrete values like part-of-speech tags, but are represented as high dimensional real-valued vectors. Our proposed method is a substantial extension of tree kernel for modeling such structures.

## 3 Representation of Tweets Propagation

On microblogging platforms, the follower/friend relationship embeds shared interests among the users. Once a user has posted a tweet, all his followers will receive the tweet. Furthermore, Twitter allows a user to retweet or comment another user's post, so that the information could reach beyond the network of the original creator.

We model the propagation of each source tweet as a tree structure $T(r) = \langle V, E \rangle$, where $r$ is the source tweet as well as the root of the tree, $V$ refers to a set of nodes each representing a responsive post (i.e., retweet or reply) of an user at a certain time to the source tweet $r$ which initiates the circulation, and $E$ is a set of directed edges corresponding to the response relation among the nodes in $V$. If there exists a directed edge from $v_i$ to $v_j$, it means $v_j$ is a direct response to $v_i$.

More specifically, each node $v \in V$ is a tuple $v = (u_v, w_v, t_v)$, which provides the following in-

formation: $u_v$ is the creator of the post, $w_v$ represents the text content of the post, and $t_v$ is the time lag between the source tweet $r$ and $v$. In our case, $u_v$ contains attributes of the user such as # of followers/friends, verification status, # of history posts, etc., $w_v$ is a vector of bag-of-words representing the post's content.

## 4 Propagation Tree Kernel Modeling

In this section, we describe our rumor detection model based on propagation trees using kernel method called Propagation Tree Kernel (PTK). Our task is, given a propagation tree $T(r)$ of a source tweet $r$, to predict the label of $r$.

### 4.1 Background of Tree Kernel

Before presenting our proposed algorithm, we briefly present the traditional tree kernel, which our PTK model is based on.

Tree kernel was designed to compute the syntactic and semantic similarity between two natural language sentences by implicitly counting the number of common subtrees between their corresponding parse trees. Given a syntactic parse tree, each node with its children is associated with a grammar production rule. Figure 3 illustrates the syntactic parse tree of "cut a tree" and its subtrees. A subtree is *defined* as any subgraph which have more than one nodes, with the restriction that entire (not partial) rule productions must be included. For example, the fragment [NP [D a]] is excluded because it contains only part of the production NP → D N (Collins and Duffy, 2001).

Following Collins and Duffy (2001), given two parse trees $T_1$ and $T_2$, the kernel function $K(T_1, T_2)$ is defined as:

$$\sum_{v_i \in V_1} \sum_{v_j \in V_2} \Delta(v_i, v_j) \qquad (1)$$

where $V_1$ and $V_2$ are the sets of all nodes respectively in $T_1$ and $T_2$, and each node is associated with a production rule, and $\Delta(v_i, v_j)$ evaluates the common subtrees rooted at $v_i$ and $v_j$. $\Delta(v_i, v_j)$ can be computed using the following recursive procedure (Collins and Duffy, 2001):

1) **if** the production rules at $v_i$ and $v_j$ are different, **then** $\Delta(v_i, v_j) = 0$;

2) **else if** the production rules at $v_i$ and $v_j$ are same, and $v_i$ and $v_j$ have only leaf children (i.e., they are pre-terminal symbols), **then** $\Delta(v_i, v_j) = \lambda$;

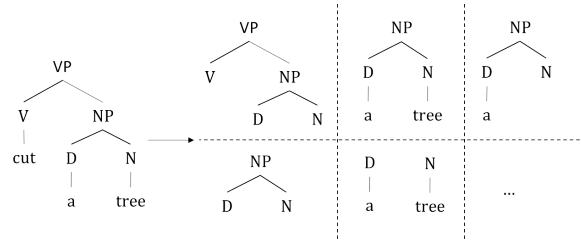

Figure 3: A syntactic parse tree and subtrees.

3) **else** $\Delta(v_i, v_j) = \lambda \prod_{k=1}^{min(nc(v_i), nc(v_j))} (1 + \Delta(ch(v_i, k), ch(v_j, k)))$.

where $nc(v)$ is the number of children of node $v$, $ch(v, k)$ is the $k$-th child of node $v$, and $\lambda$ $(0 < \lambda \le 1)$ is a decay factor. $\lambda = 1$ yields the number of common subtrees; $\lambda < 1$ down weighs the contribution of larger subtrees to make the kernel value less variable with respect to subtree size.

### 4.2 Our PTK Model

To classify propagation trees, we can calculate the similarity between the trees, which is supposed to reflect the distinction of different types of rumors and non-rumors based on structural, linguistic and temporal properties. However, existing tree kernels cannot be readily applied on propagation trees because 1) unlike parse tree where the node is represented by enumerable nominal value (e.g., part-of-speech tag), the propagation tree node is given as a vector of continuous numerical values representing the basic properties of the node; 2) the similarity of two parse trees is based on the count of common subtrees, for which the commonality of subtrees is evaluated by checking if the same production rules and the same children are associated with the nodes in two subtrees being compared, whereas in our context the similarity function should be defined softly since hardly two nodes from different trees are same.

With the representation of propagation tree, we first define a function $f$ to evaluate the similarity between two nodes $v_i$ and $v_j$ (we simplify node representation such as $v_i = (u_i, w_i, t_i)$):

$$f(v_i, v_j) = e^{-t} \left( \alpha \mathcal{E}(u_i, u_j) + (1 - \alpha) \mathcal{J}(c_i, c_j) \right)$$

where $t = |t_i - t_j|$ is the absolute difference between the time lags of $v_i$ and $v_j$, $\mathcal{E}$ and $\mathcal{J}$ are user-based similarity and content-based similarity, respectively, and $\alpha$ is the trade-off parameter. The intuition of using exponential function of $t$ to

scale down the similarity is to capture the discriminant signals or patterns at the different stages of propagation. For example, a questioning message posted very early may signal a false rumor while the same posted far later from initial post may indicate the rumor is still unverified, despite that the two messages are semantically similar.

The user-based similarity is defined as an Euclidean distance $\mathcal{E}(u_i, u_j) = ||u_i - u_j||_2$, where $u_i$ and $u_j$ are the user vectors of node $v_i$ and $v_j$ and $|| \bullet ||_2$ is the 2-norm of a vector. Here $\mathcal{E}$ is used to capture the characteristics of users participating in spreading rumors as discriminant signals, throughout the entire stage of propagation.

Contentwise, we use Jaccard coefficient to measure the similarity of post content:

$$\mathcal{J}(c_i, c_j) = \frac{|Ngram(c_i) \cap Ngram(c_j)|}{|Ngram(c_i) \cup Ngram(c_j)|}$$

where $c_i$ and $c_j$ are content words in two nodes. For n-grams here, we adopt both uni-grams and bi-grams. It can capture cue terms e.g., 'false', 'debunk', 'not true', etc. commonly occurring in rumors but not in non-rumors.

Given two propagation trees $T_1 = \langle V_1, E_1 \rangle$ and $T_2 = \langle V_2, E_2 \rangle$, PTK aims to compute the similarity between $T_1$ and $T_2$ iteratively based on enumerating all pairs of most similar subtrees. First, for each node $v_i \in V_1$, we obtain $v_i' \in V_2$, the most similar node of $v_i$ from $V_2$:

$$v_i' = \arg\max_{v_j \in V_2} f(v_i, v_j)$$

Similarly, for each $v_j \in V_2$, we obtain $v_j' \in V_1$:

$$v_j' = \arg\max_{v_i \in V_1} f(v_i, v_j)$$

Then, the propagation tree kernel $K_P(T_1, T_2)$ is defined as:

$$\sum_{v_i \in V_1} \Lambda(v_i, v_i') + \sum_{v_j \in V_2} \Lambda(v_j', v_j) \qquad (2)$$

where $\Lambda(v, v')$ evaluates the similarity of two subtrees rooted at $v$ and $v'$, which is computed recursively as follows:

1) **if** $v$ or $v'$ are leaf nodes, **then** $\Lambda(v, v') = f(v, v')$;

2) **else**
$\Lambda(v, v') = f(v, v') \prod_{k=1}^{min(nc(v), nc(v'))} (1 + \Lambda(ch(v, k), ch(v', k)))$

Note that unlike traditional tree kernel, in PTK the node similarity $f \in [0, 1]$ is used for softly counting similar subtrees instead of common subtrees. Also, $\lambda$ in tree kernel is not needed as subtree size is not an issue here thanks to node similarity $f$.

PTK aims to capture discriminant patterns from propagation trees inclusive of user, content and temporal traits, which is inspired by prior analyses on rumors spreading, e.g., user information can be a strong clue in the initial broadcast, content features are important throughout entire propagation periods, and structural and temporal patterns help for longitudinal diffusion (Zubiaga et al., 2016; Kwon et al., 2017).

### 4.3 Context-Sensitive Extension of PTK

One defect of PTK is that it ignores the clues outside the subtrees, e.g., how the information propagates from source post to the current subtree. Intuitively, propagation paths provide further clue for determining the truthfulness of information since they embed the route and context of how the propagation happens. Therefore, we propose context-sensitive TPK (cPTK) by considering the propagation paths from the root of the tree to the roots of subtrees, which shares similar intuition with the context-sensitive tree kernel (Zhou et al., 2007).

For a propagation tree node $v \in T(r)$, let $L_v^r$ be the length (i.e., # of nodes) of the propagation path from root $r$ to $v$, and $v[x]$ be the $x$-th ancestor of $v$ on the path starting from $v$ ($0 \leq x < L_v^r, v[0] = v, v[L_v^r - 1] = r$). cPTK evaluates the similarity between two trees $T_1(r_1)$ and $T_2(r_2)$ as follows:

$$\sum_{v_i \in V_1} \sum_{x=0}^{L_{v_i}^{r_1} - 1} \Lambda_x(v_i, v_i') + \sum_{v_j \in V_2} \sum_{x=0}^{L_{v_j}^{r_2} - 1} \Lambda_x(v_j', v_j)$$
$$(3)$$

where $\Lambda_x(v, v')$ measures the similarity of subtrees rooted at $v[x]$ and $v'[x]$ for context-sensitive evaluation, which is computed as follows:

1) **if** $x > 0$, $\Lambda_x(v, v') = f(v[x], v'[x])$, where $v[x]$ and $v'[x]$ are the $x$-th ancestor nodes of $v$ and $v'$ on the respective propagation path.

2) **else** $\Lambda_x(v, v') = \Lambda(v, v')$, namely PTK.

Clearly, PTK is a special case of cPTK when $x = 0$ (see equation 3). cPTK evaluates the occurrence of both context-free (without considering ancestors on propagation paths) and context-sensitive cases.

### 4.4 Rumor Detection via Kernel Learning

The advantage of kernel-based method is that we can avoid painstakingly engineering the features. This is possible because the kernel function can explore an implicit feature space when calculating the similarity between two objects (Culotta and Sorensen, 2004).

We incorporate the proposed tree kernel functions, i.e., PTK (equation 2) or cPTK (equation 3), into a supervised learning framework, for which we utilize a kernel-based SVM classifier. We treat each tree as an instance, and its similarity values with all training instances as feature space. Therefore, the kernel matrix of training set is $m \times m$ and that of test set is $n \times m$ where $m$ and $n$ are the sizes of training and test sets, respectively.

For our multi-class task, we perform a one-vs-all classification for each label and then assign the one with the highest likelihood among the four, i.e., non-rumor, false rumor, true rumor or unverified rumor. We choose this method due to interpretability of results, similar to recent work on occupational class classification (Preoţiuc-Pietro et al., 2015; Lukasik et al., 2015).

## 5 Experiments and Results

### 5.1 Data Sets

To our knowledge, there is no public dataset available for classifying propagation trees, where we need source tweets, more accurately, tree roots together with propagation structure, to be appropriately annotated.

We constructed our datasets based on a couple of reference datasets, namely Twitter15 (Liu et al., 2015) and Twitter16 (Ma et al., 2016). The original datasets were used for binary classification of rumor and non-rumor with respect to a given event that contain its relevant tweets.

First, we extracted the popular source tweets[2] that are highly retweeted or replied. We then collected all the propagation threads (i.e., retweets and replies) for these source tweets. Because Twitter API cannot retrieve the retweets or replies, we gathered the retweet users for a given tweet from Twrench[3] and crawled the replies through Twitter's web interface.

Finally, we annotated the source tweets by referring to the events they are from. We first turned

---

[2]Though unpopular tweets could be false, we ignore them as they do not draw much attention and are hardly impactful

[3]https://twren.ch

Table 1: Statistics of the datasets

| Statistic | Twitter15 | Twitter16 |
|---|---|---|
| # of users | 276,663 | 173,487 |
| # of source tweets | 1,490 | 818 |
| # of threads | 331,612 | 204,820 |
| # of non-rumors | 374 | 205 |
| # of false rumors | 370 | 205 |
| # of true rumors | 372 | 205 |
| # of unverified rumors | 374 | 203 |
| Avg. time length / tree | 1,337 Hours | 848 Hours |
| Avg. # of posts / tree | 223 | 251 |
| Max # of posts / tree | 1,768 | 2,765 |
| Min # of posts / tree | 55 | 81 |

the label of each event in Twitter15 and Twitter16 from binary to quaternary according to the veracity tag of the article in rumor debunking websites (e.g., snopes.com, Emergent.info, etc). Then we labeled the source tweets by following these rules: 1) Source tweets from unverified rumor events or non-rumor events are labeled same as event label; 2) For a source tweet in false rumor event, we flip over the label and assign true to the source tweet if it expresses denial type of stance; otherwise, the label is assigned as false; 3) The analogous flip-over/not-change rule applies to source tweets from true rumor events.

Table 1 gives statistics on the resulting datasets.

### 5.2 Experimental Setup

We compare our kernel-based method against the following baselines:

**SVM-TS**: A linear SVM classification model that uses time-series to model the variation of a set of hand-crafted features (Ma et al., 2015).

**DTR**: A Decision-Tree-based Ranking method to identify trending rumors (Zhao et al., 2015), which searches for enquiry phrases and clusters disputed factual claims, and ranked the clustered results based on statistical features.

**DTC** and **SVM-RBF:** The Twitter information credibility model using Decision Tree Classifier (Castillo et al., 2011) and the SVM-based model with RBF kernel (Yang et al., 2012), respectively, both using hand-crafted features based on the overall statistics of the posts.

**RFC:** The Random Forest Classifier proposed by Kwon et al. (2017) using three parameters to fit the temporal properties and an extensive set of hand-crafted features related to the user, linguistic and structure characteristics.

**GRU:** The RNN-based rumor detection model proposed by Ma et al. (2016) with gated recurrent

Table 2: Rumor detection results (NR: Non-Rumor; FR: False Rumor; TR: True Rumor; UR: Unverified Rumor)

(a) Twitter15 Dataset

| Method | Acc. | NR $F_1$ | FR $F_1$ | TR $F_1$ | UR $F_1$ |
|--------|------|----------|----------|----------|----------|
| DTR | 0.409 | 0.501 | 0.311 | 0.364 | 0.473 |
| SVM-RBF | 0.318 | 0.455 | 0.037 | 0.218 | 0.225 |
| DTC | 0.454 | 0.733 | 0.355 | 0.317 | 0.415 |
| SVM-TS | 0.544 | 0.796 | 0.472 | 0.404 | 0.483 |
| RFC | 0.565 | 0.810 | 0.422 | 0.401 | 0.543 |
| GRU | 0.646 | 0.792 | 0.574 | 0.608 | 0.592 |
| BOW | 0.548 | 0.564 | 0.524 | 0.582 | 0.512 |
| PTK- | 0.657 | 0.734 | 0.624 | 0.673 | 0.612 |
| cPTK- | 0.697 | 0.760 | 0.645 | 0.696 | 0.689 |
| PTK | 0.710 | **0.825** | 0.685 | 0.688 | 0.647 |
| cPTK | **0.750** | 0.804 | **0.698** | **0.765** | **0.733** |

(b) Twitter16 Dataset

| Method | Acc. | NR $F_1$ | FR $F_1$ | TR $F_1$ | UR $F_1$ |
|--------|------|----------|----------|----------|----------|
| DTR | 0.414 | 0.394 | 0.273 | 0.630 | 0.344 |
| SVM-RBF | 0.321 | 0.423 | 0.085 | 0.419 | 0.037 |
| DTC | 0.465 | 0.643 | 0.393 | 0.419 | 0.403 |
| SVM-TS | 0.574 | 0.755 | 0.420 | 0.571 | 0.526 |
| RFC | 0.585 | 0.752 | 0.415 | 0.547 | 0.563 |
| GRU | 0.633 | 0.772 | 0.489 | 0.686 | 0.593 |
| BOW | 0.585 | 0.553 | 0.556 | 0.655 | 0.578 |
| PTK- | 0.653 | 0.673 | 0.640 | 0.722 | 0.567 |
| cPTK- | 0.702 | 0.711 | 0.664 | 0.816 | 0.608 |
| PTK | 0.722 | **0.784** | 0.690 | 0.786 | 0.644 |
| cPTK | **0.732** | 0.740 | **0.709** | **0.836** | **0.686** |

unit for representation learning of high-level features from relevant posts over time.

**BOW:** A naive baseline we worked by representing the text in each tree using bag-of-words and building the rumor classifier with linear SVM.

Our models: **PTK** and **cPTK** are our full PTK and cPTK models, respectively; **PTK-** and **cPTK-** are the setting only using content while ignoring user properties.

We implemented DTC and RFC with Weka[4], SVM models with LibSVM[5] and GRU with Theano[6]. We held out 10% of the trees in each dataset for model tuning, and for the rest of the trees, we performed 3-fold cross-validation. We used accuracy, $F_1$ measure as evaluation metrics.

### 5.3 Experimental Results

Table 2 show that our proposed methods outperform all the baselines on both datasets.

---

[4] http://www.cs.waikato.ac.nz/ml/weka/
[5] https://www.csie.ntu.edu.tw/~cjlin/libsvm/
[6] http://deeplearning.net/software/theano/

Among all baselines, **GRU** performs the best, which learns the low-dimensional representation of responsive tweets by capturing the textual and temporal information. This indicates the effectiveness of complex signals indicative of rumors beyond cue words or phrases (e.g., "what?", "really?", "not sure", etc.). This also justifies the good performance of **BOW** even though it only uses uni-grams for representation. Although **DTR** uses a set of regular expressions, we found only 19.59% and 22.21% tweets in our datasets containing these expressions. That is why the results of **DTR** are not satisfactory.

**SVM-TS** and **RFC** are comparable because both of them utilize an extensive set of features especially focusing on temporal traits. But none of the models can directly incorporate structured propagation patterns for deep similarity comparison between propagation trees. **SVM-RBF**, although using a non-linear kernel, is based on traditional hand-crafted features instead of the structural kernel like ours. So, they performed obviously worse than our approach.

Representation learning methods like **GRU** cannot easily utilize complex structural information since learning features from networked data is not studied well. Our models capture complex propagation patterns from structured data rich of linguistic, user and temporal signals. The superiority of our models is clear: **PTK-** which only uses text is already better than **GRU**, demonstrating the importance of propagation structures. **PTK** that combines text and user yields better results on both datasets, implying that both properties are complementary and **PTK** integrating flat and structured information is more effective.

It is also observed that **cPTK** outperforms **PTK** except for non-rumor class. This suggests the context-sensitive modeling based on **PTK** is effective for different types of rumors, but for non-rumors, it seems that considering context of propagation path is not always helpful. This might be due to the generally weak signal from node properties on the paths during non-rumor's diffusion as user distribution patterns in non-rumors are not as obvious as in rumors. This is not an issue in **cPTK-** since user information is not considered. Over all classes, **cPTK** achieves the highest accuracies on both datasets.

Furthermore, we observe that all the baseline methods perform much better on non-rumors than

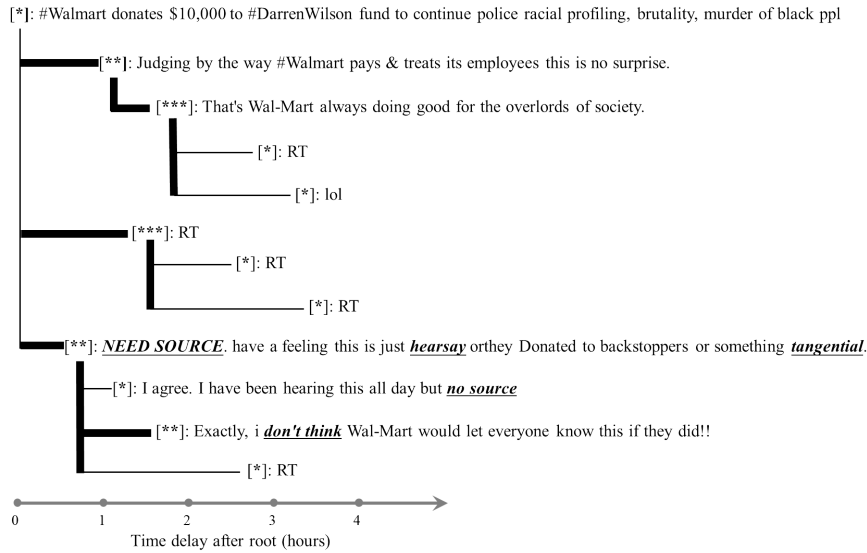

Figure 4: The example subtree of a rumor captured by the algorithm at early stage of propagation

on rumors. This is because the exiting features were defined for binary (rumor vs. non-rumor) classification in previous works. So, they do not perform well for finer-grained classes. Our approach can differentiate various classes much better by deep, detailed comparison of different patterns based on propagation structure.

### 5.4 Early Detection Performance

Detecting rumors at an early stage of propagation is very important so that preventive measures could be taken. In early detection task, all the posts after a detection deadline are invisible during test. The earlier the deadline, the less propagation information can be available.

Figure 5 shows the performances of our **PTK** and **cPTK** models versus **RFC** (the best system based on feature engineering), **GRU** (the best system based on RNN) and **DTR** (an early-detection-specific algorithm) against various deadlines. In the first few hours, our approach demonstrates superior early detection performance than other models. Particularly, **cPTK** achieve 75% accuracy on Twitter15 and 73% on Twitter16 within 24 hours, that is much faster than other models.

Our analysis shows that rumors typically demonstrate more complex propagation substructures especially at early stage. Figure 4 shows a detected subtree of a false rumor spread in its first few hours, where influential users are somehow captured to boost its propagation and the information flows among the users with an obvious unpopular-to-popular-to-unpopular trend in terms

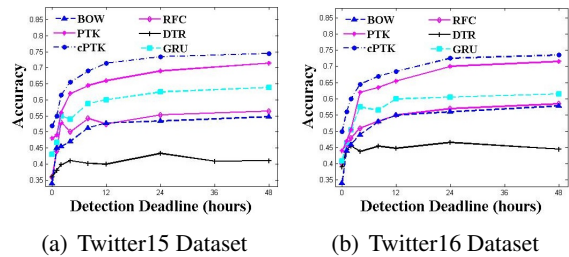

(a) Twitter15 Dataset (b) Twitter16 Dataset

Figure 5: Results of rumor early detection

of user popularity, but such pattern was not witnessed in non-rumors in early stage. Many textual signals (underlined) can also be observed in that early period. Our method can learn such structures and patterns naturally, but it is difficult to know and hand-craft them in feature engineering.

### 6 Conclusion

We propose a novel approach for detecting rumors in microblog posts based on kernel learning method using propagation trees. A propagation tree encodes the spread of a hypothesis (i.e., a source tweet) with complex structured patterns and flat information regarding content, user and time associated with the tree nodes. Enlightened by tree kernel techniques, our kernel method learns discriminant clues for identifying rumors of finer-grained levels by directly measuring the similarity among propagation trees via kernel functions. Experiments show that our approach outperforms state-of-the-art baselines with large margin for both general and early rumor detection tasks.

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
