# Peer review of "Detect Rumors in Microblog Posts Using Propagation Structure via Kernel Learning"

_ACL 2017 — decision unknown_

[Official Review · Reviewer 1 · rating 4 · confidence 4]
soundness 3 · originality 4 · clarity 4 · impact 4 · substance 4 · appropriateness 5 · meaningful comparison 4 · presentation format Oral Presentation

- Strengths:

The authors propose a kernel-based method that captures high-order patterns
differentiting different types of rumors by evaluating the similarities between
their propagation tree structures.

- Weaknesses:

maybe the maths is not always clear in Sect. 4. 

- General Discussion:

The authors propose a propagation tree kernel, a kernel-based method that
captures high-order patterns differentiating types of rumors by evaluating the
similarities between their propagation tree structures. The proposed approach
detects rumors more quickly and with a higher accuracy compared to the one
obtained by the state of the art methods.

The data set should be made public for research purposes.

Typos need to be fixed (e.g. 326/3277: any subgraph which have->has; 472:
TPK->PTK; 644: Table 2 show+s), missing information needs to be added (875:
where was it published?), information needs to be in the same format (e.g. 822
vs 897). Figure 5 is a bit small.